# Sex Differences in the Footprint Analysis During the Entire Gait Cycle in a Functional Equinus Condition: Novel Cross Sectional Research

**Eva María Martínez-Jiménez** [1], **Ricardo Becerro de Bengoa-Vallejo** [1],
**Marta Elena Losa-Iglesias** [2], **José Ignacio Díaz-Velázquez** [3], **Israel Casado-Hernández** [1],
**Cesar Calvo-Lobo** [1,*], **Daniel López-López** [4] and **David Rodríguez-Sanz** [1]

[1]  Facultad de Enfermería, Fisioterapia y Podología, Universidad Complutense de Madrid, 28040 Madrid, Spain
[2]  Faculty of Health Sciences, Universidad Rey Juan Carlos, 28922 Alcorcón, Spain
[3]  Departament of Physiotherapy and Podiatry, Premium Madrid, 28017 Madrid, Spain
[4]  Research, Health and Podiatry Unit, Department of Health Sciences, Faculty of Nursing and Podiatry, Universidade da Coruña, 15403 Ferrol, Spain
*  Correspondence: cecalvo19@hotmail.com; Tel.: +34-913-941-532



**Featured Application: Different shoes may be necessary according to sex in functional equinus condition. The first step was the most reliable step to register in clinical practice and research**

**Abstract:** Some studies suggest that gender is related to gait. Females show significantly higher ankle motion and vertical ground reaction forces. Males have significantly larger plantar contact surface areas in all regions of the foot than females in most, but not all, prior studies. However, there is no research on sex differences in a functional equinus condition. In this study, 119 individuals, including 59 females (29.7 ± 5.15 years, 58.74 ± 6.66 kg, 163.65 ± 5.58 cm) and 60 males (31.22 ± 6.06 years, 75.67 ± 9.81 kg, 177.10 ± 6.16 cm), with a functional equinus condition walked onto a pressure platform. In two separate testing sessions, five trials of each foot were conducted for the first, second, and third steps. We measured the contact surface areas for each of the three phases of the stance phase. We computed the intraclass correlation coefficient and standard error of the mean to assess the reliability. We found significantly greater contact surface areas in males than females in the first, second, and third steps in all phases of the stance phase: heel strike, mid-stance, and take-off. This is important information for the design of footwear and orthotics and gender knowledge. In a functional equinus condition, males have registered greater contact surface areas than females in all phases of the dynamic footprint of the stance phase.

**Keywords:** gender; foot; reliability; platform; biomechanical phase

---

## 1. Introduction

Different habits (such as frequency of sport activity and shoe wearing practices), and individual characteristics (such as sex, body mass index, and age), are related to adult foot morphology [1]. Human foot shape also differs among ethnic groups [2] and changes over the course of postnatal development [3].

Anthropometric studies have shown considerable sex differences in the foot bones [4]. In three-dimensional analyses, sex was associated with ankle width, Achilles tendon width, and heel width [5]. In contrast to males, females have greater generalized joint laxity following the onset of puberty [6]. Postpubescent athletes had greater knee anterior/posterior forces, as well as mediolateral resultant forces, after jumping [7].

Research on footprint analysis during walking in two-dimensional analyses of data from an in-shoe pressure measurement system has been carried out. Sex was not significantly related to the peak pressure, contact time, pressure-time integral, and instant of peak pressure. The force-time integral, however, was significantly greater in males than females under the first, third, and fourth metatarsal heads [8]. The maximum force was also significantly higher in males under the heel [8] and the first and third metatarsal heads, and the mean force was greater in males under the third metatarsal head [8]. It should be noted that none of these variables were normalized to body weight or height. Kandil et al. [9] studied the plantar pressures in a standing position and found that females had significantly higher pressures in the heel than males. Several authors [8,10,11] observed that males had significantly larger contact areas in all regions of the foot than females. Other authors, in a sequence of footprints during a single gait cycle, called the dynamic footprint [12], found no differences [13].

A functional equinus condition can be defined as "dorsiflexion limitation of the ankle with the knee extended/flexed (excluding osseous restriction)" [13,14]. A functional equinus condition is a non-symptomatic condition, but its incidence is high. In a study of 209 consecutive patients with musculoskeletal problems in the foot, a prevalence of 96.5% of this condition was found and has been linked to foot problems including plantar fasciitis and metatarsalgia [15]. It may promote clinical alterations in the Achilles tendon and triceps surae muscle [13]. Equinus is significantly related to lower limb injuries (e.g., anterior cruciate ligament rupture), asymmetric loading patterns, and alterations in triceps surae contraction [16]. In footwear, a prolonged or chronic use of high-heeled shoes can cause the Achilles tendon to become stiff and rigid [17]. A lack of adequate ankle dorsiflexion can result in compensation within the gait cycle, such as a near heel lift and an increase in forefoot pressures [18] Once the ankle is restricted, the midtarsal joint is the next joint through which dorsiflexion may occur [19]. This is achieved by excessive pronation of the foot [19]. The pronation of the midtarsal joint of the functional equinus condition and the postpubescent laxity of women are two factors that could increase the variable of the surface of the footprint. Therefore, it is necessary to check if ligamentous laxity is an agent that provides sex differences in this pathology.

Gender differences in plantar footprint variables with functional Equinus have not been compared with a biomechanical approach for the first, second, and third steps. If they exist, they should be taken into consideration for the specific design of footwear by sex, especially in those situations with higher ground contact forces, such as sport. Our measurements were focused on the three stages of the stance phase: heel strike, mid-stance, and take off, in order to check sex differences. The main purpose of this study was to compare males and females in terms of surface variables for the first, second, and third walking steps with a pressure platform.

## 2. Materials and Methods

### 2.1. Subjects

One hundred and nineteen uninjured subjects (59 women and 60 men) participated [5]. All of them were European Caucasians. The Ethics Committee of Universidad Rey Juan Carlos approved the study, and all subjects gave their written informed consent before participating. Gender, age, foot size, height, and weight were recorded (Table 1). Ethical standards in human experimentation contained in the World Medical Association Declaration of Helsinki, the Council of Europe Convention on Human Rights and Biomedicine, the UNESCO Universal Declaration on the Human Genome and Human Rights, and those of the relevant national bodies and institutions were observed at all times.

The inclusion criteria were as follows: age between 18 and 40 years; a normal or overweight body mass index (between 18.5 or less than 30, based on Quetelet's equation of BMI = $weight/height^2$); and the ability to walk independently, with no previous lower limb injury. All individuals had a functional equinus condition. Gastrocnemius soleus equinus is the inability of the ankle to dorsiflex beyond a neutral position with the knee extended (it remains $<0°$) [13,14,20] or with the knee flexed (it remains $<0°$) [13,14,20]. Both men and women have an inability of dorsiflex between −10 to −15 degrees. Values

greater than −15° were excluded from the study. The subjects did not practice sports and their usual shoes had a 3 to 5 cm heel.

**Table 1.** Clinical characteristics of the sample population.

| | Total Group n = 119 (100%) Mean (SD) | Female Group n = 59 (49.6%) Mean (SD) | Male Group n = 60 (50.4%) Mean (SD) | *p*-Value * |
|---|---|---|---|---|
| Age (years) | 30.50 (3.65) | 29.76 (5.15) | 31.22 (6.06) | *p* = 0.161 |
| Weight (kg) | 67.28 (11.92) | 58.74 (6.66) | 75.67 (9.81) | *p* = 0.001 |
| Height (cm) | 170.43 (8.94) | 163.65 (5.58) | 177.10 (6.16) | *p* = 0.001 |
| Shoe size (Eu) | 40.67 (2.68) | 38.46 (1.21) | 42.85 (1.78) | *p* = 0.001 |

Abbreviations: kg, kilograms; cm, centimeters; BMI, body mass index; SD, standard deviation; CI 95%, confidence interval 95%. In all the analyses, $p < 0.05$ (with a 95% confidence interval) was considered statistically significant. *p*-values are from a Student's *t*-test *.

Gastrocnemius soleus equinus was assessed with the knee extended and flexed. The amount of ankle dorsiflexion was determined by using a goniometer to measure the angle between the plantar aspect of the heel (medially or laterally) and the tibia. Care was taken to maintain the subtalar joint in a neutral position and to measure ankle dorsiflexion and not midfoot dorsiflexion (rocker bottom) or midfoot equinus (pseudoequinus) [15,20]. The Silfverskiöld test was performed to differentiate gastrocnemius equinus from other types of equinus [15,21]. A normal amount of ankle dorsiflexion is approximately 108 with the knee extended and 208 with the knee flexed [12].

The exclusion criteria were determined with a questionnaire, and were as follows: obesity (more than 30 based on Quetelet's equation of BMI = weight/height$^2$); history of problems with the feet or lower extremities or any pathological condition in the past 12 months [22,23]; having a history of foot surgery; congenital or acquired deformity of the foot (flat feet, cavus foot, hallux valgus, hallux limitux, hammer toes, congenital, or traumatic deformity of the lower limb [22–24]), to have normal dorsiflexion in the ankle joint complex [21], and the presence of musculoskeletal and joint injuries, pelvic pain, ankle sprains, and lower back pain [25] identified during clinical examination; visual and/or hearing impairments; and any problems in the lower limbs or spine that might affect the normal gait.

*2.2. Procedures*

The pressure platform was located in the center of a corridor that was 6.4 m long and was flush with the floor surface. Subjects performed the first, second, and third steps barefoot [15,23]. Subjects were instructed not to look at the platform or the ground to protect the reliability of the measures [23]. We randomly assigned the order of the steps (first, second, and third) [16,26–30] and the lower limb (left vs. right) to be assessed. Four or five tests were performed on each protocol to familiarize the subject with the procedure and to determine his or her starting position. These tests ensured a successful arrival of the full foot on the platform. A comfortable walking speed was chosen by the subjects [22,30,31]. After reaching the platform, the subjects continued to take a minimum of 3 to 4 steps [30]. We excluded trials in which the subject's entire foot did not make contact with the platform [8]. We recorded measurements in five successful trials per leg and step combination (30 trials in all per subject in a session).

We repeated the same data collection procedures in a second session at the same hour of the day one week after the first session, with the objective of checking the reliability of the data collected in both weeks and demonstrating that the test is reproducible. We made a new randomization of steps and limbs for this session. Subjects wore the same shoes on session days (although they participated barefoot).

The platform was made with capacitive sensors. The dielectric material of which they consisted has excellent elastic properties so can return to the original position after use. After each trial, we waited 30 s to allow the platform material to return to its original state [31]. The Podoprint (Medicapteurs; Balma, France) platform we used has an active area of 400 mm × 400 mm, with 2304 sensors [31].

Technical specifications of the pressure platform are shown in the Table 2. This pressure platform is indicated for use in studies of footprints during the stance phase of the gait cycle [21,32,33]. We used the platform in self-calibrating mode. We used the manufacturer's software to analyze the data.

**Table 2.** Intra-class correlation coefficient (ICC) and standard errors of the mean (SEM) for Contact Surface Areas, Females.

| Protocol, Foot, and Biomechanical Phase | FIRST WEEK | | | SECOND WEEK | | | BOTH WEEKS | | |
| | 95% CI | | | 95% CI | | | 95% CI | | |
| | ICC (Lower Limit–Upper Limit) | SEM | | ICC (Lower Limit–Upper Limit) | SEM | | ICC (Lower Limit–Upper Limit) | SEM | |
|---|---|---|---|---|---|---|---|---|---|
| First step, Left foot, 20% | 0.600(0.489–0.708) | 8.18 | | 0.674(0.573–0.767) | 7.30 | | 0.809(0.698–0.882) | 5.34 | |
| First step, Right foot, 20% | 0.555(0.440–0.671) | 8.58 | | 0.507(0.389–0.629) | 7.50 | | 0.724(0.575–0.826) | 5.76 | |
| First step, Left foot, 35% | 0.691(0.594–0.781) | 6.79 | | 0.737(0.649–0.816) | 6.29 | | 0.888(0.819–0.932) | 3.98 | |
| First step, Right foot, 35% | 0.653(0.549–0.751) | 6.72 | | 0.725(0.633–0.807) | 6.01 | | 0.875(0.798–0.924) | 3.91 | |
| First step, Left foot, 92% | 0.357(0.239–0.492) | 5.29 | | 0.472(0.352–0.598) | 4.96 | | 0.611(0.423–0.749) | 3.76 | |
| First step, Right foot, 92% | 0.400(0.280–0.532) | 5.58 | | 0.365(0.246–0.499) | 4.78 | | 0.611(0.422–0.749) | 3.71 | |
| Second step, Left foot, 20% | 0.655(0.552–0.753) | 7.53 | | 0.629(0.521–0.732) | 6.99 | | 0.797(0.681–0.874) | 5.19 | |
| Second step, Right foot, 20% | 0.629(0.522–0.732) | 7.75 | | 0.601(0.490–0.709) | 7.49 | | 0.783(0.660–0.865) | 5.41 | |
| Second step, Left foot, 35% | 0.769(0.688–0.840) | 5.45 | | 0.663(0.560–0.759) | 5.68 | | 0.879(0.805–0.927) | 3.57 | |
| Second step, Right foot, 35% | 0.752(0.667–0.827) | 5.35 | | 0.716(0.623–0.800) | 5.72 | | 0.848(0.757–0.907) | 4.03 | |
| Second step, Left foot, 92% | 0.224(0.117–0.357) | 4.93 | | 0.337(0.220–0.472) | 4.96 | | 0.653(0.478–0.778) | 3.13 | |
| Second step, Right foot, 92% | 0.324(0.207–0.461) | 5.38 | | 0.392(0.273–0.525) | 4.79 | | 0.681(0.514–0.798) | 3.23 | |
| Third step, Left foot, 20% | 0.655(0.551–0.753) | 6.63 | | 0.692(0.594–0.781) | 6.63 | | 0.791(0.672–0.870) | 5.02 | |
| Third step, Right foot, 20% | 0.599(0.488–0.708) | 7.90 | | 0.739(0.651–0.818) | 6.38 | | 0.726(0.578–0.827) | 6.07 | |
| Third step, Left foot, 35% | 0.742(0.655–0.820) | 5.84 | | 0.765(0.683–0.837) | 5.74 | | 0.885(0.814–0.930) | 3.84 | |
| Third step, Right foot, 35% | 0.762(0.679–0.835) | 5.66 | | 0.741(0.653–0.819) | 5.55 | | 0.870(0.790–0.920) | 3.93 | |
| Third step, Left foot, 92% | 0.267(0.156–0.402) | 4.67 | | 0.351(0.234–0.486) | 4.96 | | 0.678(0.512–0.795) | 3.02 | |
| Third step, Right foot, 92% | 0.304(0.189–0.440) | 4.79 | | 0.180(0.079–0.310) | 4.67 | | 0.478(0.255–0.653) | 3.53 | |

## 2.3. Variables

We extracted the surface variable at different points in the stance phase of the gait. We divided each entire dynamically registered plantar footprint into each of the biomechanical intraphases, as affirmed by Cornwall and Mc Poil [34]. They defined the Velocity-Time graph of the center of pressures with the platform during the stance phase of the gait. Characteristically, there is a triple peak pattern graph that is biomechanically related to a specific moment of the support phase: heel contact (from the beginning to 20% of the duration of the stance phase), midstance (from 20% to 35% of the duration of the stance phase), and take-off (from 35% to 92% of the duration of the stance phase; from 92% to 100%, only the toes are in contact with the ground). Therefore, 20% of the total time of the stance phase was used to record the contact surface, and we did the same at 35% and 92%. The area recorded during the entire stance phase was also analyzed and was called the global footprint. We used the end of each phase of the stance phase because it is the moment where greater velocities of the center of pressures take place [34], in order to check the influence of the ligamentous laxity.

We carried out a Kolmogorov–Smirnov test for normality assessment, and we considered a normal distribution if $p > 0.05$. We performed descriptive statistical analyses, using the mean ± standard deviation and a 95% confidence interval. For each step, leg, and stage of gait combination, we computed the intra-class correlation coefficient (ICC) to assess the reliability of each parameter, as the degree to which individuals maintain their position or value in repeated measures, as proposed Bruton, Conway and Holgate [35]. To interpret ICC values, we used the benchmarks proposed by Landis and Koch [36]:



0.20 or less, slight agreement; 0.21 to 0.40, fair; 0.41 to 0.60, moderate; 0.61 to 0.80, substantial; and 0.81 or greater, almost perfect.

We calculated standard errors of the mean (SEM) to measure the range of error in each parameter. We computed the SEM as $s_x.\sqrt{(1-r_{xx})}$, where $s_x$ is the standard deviation of the observed set of test scores, and $r_{xx}$ is the reliability coefficient, which, in our case, is the ICC. Additionally, we performed Student's t tests to compare males' and females' means for each parameter. We considered a *p* value < 0.05 as statistically significant for all tests. We used SPSS for Windows, version 20.0 (SPSS Inc., Chicago, IL, USA), for all statistical analyses.

## 3. Results

### 3.1. Characteristics of the Sample

All the variables showed a normal distribution (*p* > 0.05) by the Kolmogorov–Smirnov test. Table 1 shows the characteristics of the obtained sample. Significant differences between men and women in terms of weight, height, and shoe size are evident. There was no significant difference in age between men and women.

### 3.2. Reliability

For women, all contact surface area variables except one (right foot, third step, take off) displayed substantial to almost perfect reliability when data were aggregated across sessions (Table 2). For men, all contact surface area variables showed substantial to almost perfect reliability when data were aggregated across sessions (Table 3). Reliability was generally lower when measurements from only the first or second session were considered. The take-off phase measurements tended to have a lower reliability than measurements during heel contact or the mid-phase.

**Table 3.** Intra-class correlation coefficient (ICC) and standard errors of the mean (SEM) for Contact Surface Areas, Males.

| Protocol, Foot, and Biomechanical Phase | FIRST WEEK | | | SECOND WEEK | | | BOTH WEEKS | | |
|---|---|---|---|---|---|---|---|---|---|
| | 95% CI | | | 95% CI | | | 95% CI | | |
| | ICC (Lower Limit–Upper Limit) | | SEM | ICC (Lower Limit–Upper Limit) | | SEM | ICC (Lower Limit–Upper Limit) | | SEM |
| First step, Left foot, 20% | 0.656(0.553–0.753) | | 9.15 | 0.593(0.482–0.70)1 | | 9.59 | 0.860(0.777–0.914) | | 5.53 |
| First step, Right foot, 20% | 0.648(0.544–0.746) | | 9.60 | 0.658(0.556–0.754) | | 9.01 | 0.798(0.684–0.874) | | 6.73 |
| First step, Left foot, 35% | 0.665(0.563–0.760) | | 8.73 | 0.811(0.741–0.870) | | 6.42 | 0.891(0.824–0.934) | | 4.79 |
| First step, Right foot, 35% | 0.704(0.610–0.791) | | 7.36 | (0.793(0.719–0.857) | | 6.55 | 0.890(0.823–0.933) | | 4.50 |
| First step, Left foot, 92% | 0.542(0.427–0.659) | | 5.65 | 0.509(0.392–0.630) | | 5.62 | 0.795(0.679–0.872) | | 3.51 |
| First step, Right foot, 92% | 0.437(0.318–0.566) | | 6.00 | 0.473(0.355–0.599) | | 5.77 | 0.779(0.655–0.862) | | 3.54 |
| Second step, Left foot, 20% | 0.654(0.551–0.751) | | 8.76 | 0.778(0.700–0.846) | | 7.82 | 0.898(0.834–0.938) | | 4.91 |
| Second step, Right foot, 20% | 0.678(0.578–0.770) | | 8.46 | 0.734(0.645–0.813) | | 7.31 | 0.861(0.777–0.914) | | 5.23 |
| Second step, Left foot, 35% | 0.782(0.704–0.849) | | 6.29 | 0.779(0.701–0.847) | | 6.48 | 0.914(0.860–0.948) | | 3.91 |
| Second step, Right foot, 35% | 0.802(0.730–0.864) | | 5.94 | 0.804(0.733–0.865) | | 5.26 | 0.932(0.888–0.959) | | 3.24 |
| Second step, Left foot, 92% | 0.417(0.299–0.548) | | 5.15 | 0.420(0.301–0.550) | | 5.57 | 0.651(0.478–0.776) | | 3.78 |
| Second step, Right foot, 92% | 0.482(0.364–0.607) | | 5.84 | 0.410(0.291–0.541) | | 5.77 | 0.750(0.613–0.842) | | 3.66 |
| Third step, Left foot, 20% | 0.734(0.645–0.813) | | 7.74 | 0.747(0.661–0.823) | | 7.55 | 0.858(0.773–0.913) | | 5.45 |
| Third step, Right foot, 20% | 0.647(0.543–0.746) | | 8.95 | 0.719(0.627–0.802) | | 7.48 | 0.864(0.783–0.917) | | 5.20 |
| Third step, Left foot, 35% | 0.827(0.761–0.882) | | 5.52 | 0.812(0.742–0.871) | | 5.94 | 0.954(0.925–0.972) | | 2.86 |
| Third step, Right foot, 35% | 0.781(0.702–0.849) | | 5.92 | 0.834(0.771–0.887) | | 5.26 | 0.920(0.869–0.951) | | 3.55 |
| Third step, Left foot, 92% | 0.456(0.337–0.583) | | 5.81 | 0.397(0.279–0.529) | | 5.42 | 0.706(0.552–0.813) | | 3.73 |
| Third step, Right foot, 92% | 0.480(0.361–0.604) | | 6.04 | 0.456(0.336–0.584) | | 5.65 | 0.58(0.385–0.727) | | 4.93 |

## 3.3. Comparisons between Women and Men

Tables 4–6 show the results for comparisons between women and men for each foot, biomechanical intraphase, and session for the first, second, and third steps, respectively. During heel strike, mid-stance, and take off of the stance phase, men had significantly larger contact surface areas than women. In Figure 1, samples of each phase of the third step can be seen.

**Table 4.** Comparison of Women and Men for Contact Surface Areas, First Step.

| Biomechanical Intraphase | | | Total<br>n = 119 (100%)<br>Mean SD<br>cm$^2$ | Women<br>n = 59 (49.6%)<br>Mean SD<br>cm$^2$ | Men<br>n = 60 (50.4%)<br>Mean SD<br>cm$^2$ | Difference between<br>Women and Men<br>*p*-Value |
|---|---|---|---|---|---|---|
| First week | Left foot | 20% | 65.73 (15.46) | 59.79 (12.93) | 71.57 (15.60) | *p* = 0.001 * |
| | | 35% | 95.08 (15.99) | 86.76 (12.22) | 103.27 (15.09) | *p* = 0.001 * |
| | | 92% | 49.18 (8.22) | 45.81 (6.60) | 52.49 (8.35) | *p* = 0.001 * |
| | | Global | 148.12 (20.19) | 136.39 (13.79) | 159.65 (18.86) | *p* = 0.001 * |
| | Right foot | 20% | 64.21 (15.29) | 59.56 (12.86) | 68.79 (16.18) | *p* = 0.001 * |
| | | 35% | 95.14 (14.60) | 87.51 (11.40) | 102.64 (13.53) | *p* = 0.001 * |
| | | 92% | 49.72 (8.48) | 45.89 (7.20) | 53.48 (8.00) | *p* = 0.001 * |
| | | Global | 147.22 (19.04) | 135.69 (14.48) | 158.55 (15.99) | *p* = 0.001 * |
| Second week | Left foot | 20% | 64.95 (15.24) | 58.68 (12.79) | 71.13 (15.03) | *p* = 0.001 * |
| | | 35% | 93.58 (15.59) | 85.80 (12.27) | 101.23 (14.76) | *p* = 0.001 * |
| | | 92% | 48.82 (7.93) | 46.02 (6.83) | 51.57 (8.02) | *p* = 0.001 * |
| | | Global | 147.73 (20.99) | 135.21 (814.94) | 159.84 (18.85) | *p* = 0.001 * |
| | Right foot | 20% | 64.82 (14.13) | 59.79 (10.68) | 69.76 (15.40) | *p* = 0.001 * |
| | | 35% | 94.63 (15.17) | 86.73 (11.46) | 102.39 (14.40) | *p* = 0.001 * |
| | | 92% | 48.92 (7.80) | 45.52 (6.00) | 52.28 (7.95) | *p* = 0.001 * |
| | | Global | 146.94 (18.78) | 135.19 (14.07) | 158.69 (15.24) | *p* = 0.001 * |
| Both weeks | Left foot | 20% | 65.34 (14.82) | 59.24 (12.23) | 71.35 (14.77) | *p* = 0.001 * |
| | | 35% | 94.33 (15.46) | 86.28 (11.90) | 102.25 (14.51) | *p* = 0.001 * |
| | | 92% | 49.00 (7.57) | 45.92 (6.03) | 52.03 (7.75) | *p* = 0.001 * |
| | | Global | 147.97 (20.35) | 135.78 (13.99) | 159.75 (18.57) | *p* = 0.001 * |
| | Right foot | 20% | 64.52 (13.95) | 59.68 (10.97) | 69.28 (14.98) | *p* = 0.001 * |
| | | 35% | 94.88 (14.57) | 87.12 (11.07) | 102.52 (13.58) | *p* = 0.001 * |
| | | 92% | 49.32 (7.66) | 45.71 (5.95) | 52.88 (7.52) | *p* = 0.001 * |
| | | Global | 146.89 (18.54) | 135.44 (14.04) | 158.33 (15.17) | *p* = 0.001 * |

* Significant differences, T-Student's test (*p* > 0.05).

**Table 5.** Comparison of Women and Men for Contact Surface Areas, Second Step.

| Biomechanical Intraphase | | | Total<br>n = 119 (100%)<br>Mean SD<br>cm$^2$ | Women<br>n = 59 (49.6%)<br>Mean SD<br>cm$^2$ | Men<br>n = 60 (50.4%)<br>Mean SD<br>cm$^2$ | Difference<br>between Women<br>and Men<br>*p*-Value |
|---|---|---|---|---|---|---|
| First week | Left foot | 20% | 72.36 (15.14) | 66.22 (12.82) | 78.40 (14.90) | *p* = 0.001 * |
| | | 35% | 96.24 (14.61) | 88.50 (11.34) | 103.85 (13.48) | *p* = 0.001 * |
| | | 92% | 48.12 (6.72) | 45.48 (5.60) | 50.71 (6.75) | *p* = 0.001 * |
| | | Global | 148.09 (19.67) | 136.85 (13.90) | 159.13 (18.27) | *p* = 0.001 * |
| | Right foot | 20% | 71.13 (15.32) | 64.46 (12.72) | 77.69 (14.90) | *p* = 0.001 * |
| | | 35% | 96.24 (14.28) | 88.57 (10.75) | 103.77 (13.34) | *p* = 0.001 * |
| | | 92% | 48.29 (7.81) | 45.61 (6.54) | 50.88 (8.11) | *p* = 0.001 * |
| | | Global | 148.13 (19.23) | 135.76 (14.09) | 160.29 (15.52) | *p* = 0.001 * |
| Second week | Left foot | 20% | 74.47 (15.60) | 68.05 (11.47) | 80.78 (16.60) | *p* = 0.001 * |
| | | 35% | 96.65 (14.08) | 89.13 (9.79) | 104.04 (13.78) | *p* = 0.001 * |
| | | 92% | 48.13 (7.39) | 45.18 (6.09) | 51.38 (7.32) | *p* = 0.001 * |
| | | Global | 149.22 (19.89) | 137.29 (13.21) | 160.95 (18.37) | *p* = 0.001 * |
| | Right foot | 20% | 72.73 (14.62) | 66.04 (11.85) | 79.32 (14.17) | *p* = 0.001 * |
| | | 35% | 96.22 (13.78) | 88.27 (10.73) | 104.03 (11.89) | *p* = 0.001 * |
| | | 92% | 47.84 (7.84) | 43.99 (6.14) | 51.62 (7.51) | *p* = 0.001 * |
| | | Global | 148.39 (19.73) | 135.75 (13.16) | 160.83 (17.07) | *p* = 0.001 * |

**Table 5.** *Cont.*

| Biomechanical Intraphase | | | Total<br>n = 119 (100%)<br>Mean SD<br>cm² | Women<br>n = 59 (49.6%)<br>Mean SD<br>cm² | Men<br>n = 60 (50.4%)<br>Mean SD<br>cm² | Difference<br>between Women<br>and Men<br>*p*-Value |
|---|---|---|---|---|---|---|
| Both weeks | Left foot | 20% | 73.42 (14.92) | 67.14 (11.53) | 79.59 (15.37) | *p* = 0.001 * |
| | | 35% | 96.45 (14.09) | 88.82 (10.27) | 103.95 (13.34) | *p* = 0.001 * |
| | | 92% | 48.21 (6.53) | 45.33 (5.32) | 51.04 (6.40) | *p* = 0.001 * |
| | | Global | 148.65 (19.38) | 137.07 (13.26) | 160.04 (17.65) | *p* = 0.001 * |
| | Right foot | 20% | 71.93 (14.45) | 65.24 (11.61) | 78.51 (14.02) | *p* = 0.001 * |
| | | 35% | 96.23 (13.78) | 88.42 (10.33) | 103.90 (12.42) | *p* = 0.001 * |
| | | 92% | 48.14 (7.28) | 44.92 (5.72) | 51.25 (7.31) | *p* = 0.001 * |
| | | Global | 148.26 (19.21) | 135.75 (13.37) | 160.56 (15.86) | *p* = 0.001 * |

* Significant differences, T-Student´s test (*p* > 0.05).

**Table 6.** Comparison of Women and Men for Contact Surface Areas, Third Step.

| Biomechanical Intraphase | | | Total<br>n = 119 (100%)<br>Mean SD<br>cm² | Women<br>n = 59 (49.6%)<br>Mean SD<br>cm² | Men<br>n = 60 (50.4%)<br>Mean SD<br>cm² | Difference<br>between Women<br>and Men<br>*p*-Value |
|---|---|---|---|---|---|---|
| First week | Left foot | 20% | 75.05 (14.75) | 68.50 (11.29) | 81.48 (15.01) | *p* = 0.001 * |
| | | 35% | 97.54 (14.59) | 89.78 (11.59) | 105.18 (13.27) | *p* = 0.001 * |
| | | 92% | 48.06 (7.39) | 45.06 (5.45) | 51.01 (7.88) | *p* = 0.001 * |
| | | Global | 148.64 (19.32) | 136.69 (12.71) | 160.38 (17.45) | *p* = 0.001 * |
| | Right foot | 20% | 74.19 (15.92) | 66.18 (12.47) | 82.06 (15.06) | *p* = 0.001 * |
| | | 35% | 96.69 (14.37) | 88.96 (11.60) | 104.42 (12.66) | *p* = 0.001 * |
| | | 92% | 48.06 (7.61) | 45.48 (5.74) | 50.60 (8.38) | *p* = 0.001 * |
| | | Global | 147.61 (19.53) | 134.94 (13.90) | 160.07 (17.95) | *p* = 0.001 * |
| Second week | Left foot | 20% | 76.42 (15.58) | 68.63 (11.94) | 84.07 (15.02) | *p* = 0.001 * |
| | | 35% | 97.99 (15.25) | 89.59 (11.84) | 106.25 (13.69) | *p* = 0.001 * |
| | | 92% | 48.01 (7.41) | 44.54 (6.16) | 51.42 (6.98) | *p* = 0.001 * |
| | | Global | 148.36 (18.94) | 136.49 (14.03) | 160.03 (15.65) | *p* = 0.001 * |
| | Right foot | 20% | 75.10 (15.22) | 67.63 (12.49) | 82.45 (14.12) | *p* = 0.001 * |
| | | 35% | 97.36 (14.64) | 88.81 (10.91) | 105.77 (12.91) | *p* = 0.001 * |
| | | 92% | 47.92 (7.06) | 45.19 (5.16) | 50.66 (7.66) | *p* = 0.001 * |
| | | Global | 148.69 (19.93) | 136.19 (13.37) | 160.97 (17.58) | *p* = 0.001 * |
| Both weeks | Left foot | 20% | 75.73 (14.66) | 68.57 (10.99) | 82.77 (14.47) | *p* = 0.001 * |
| | | 35% | 97.76 (14.72) | 89.68 (11.33) | 105.71 (13.33) | *p* = 0.001 * |
| | | 92% | 48.03 (6.92) | 44.80 (5.33) | 51.21 (6.87) | *p* = 0.001 * |
| | | Global | 148.29 (18.77) | 136.46 (13.05) | 159.91 (16.12) | *p* = 0.001 * |
| | Right foot | 20% | 74.65 (14.99) | 66.91 (11.60) | 82.25 (14.09) | *p* = 0.001 * |
| | | 35% | 96.96 (14.24) | 88.88 (10.89) | 105.03 (12.56) | *p* = 0.001 * |
| | | 92% | 48.03 (6.92) | 45.33 (4.89) | 50.73 (7.61) | *p* = 0.001 * |
| | | Global | 148.15 (19.43) | 135.56 (13.06) | 160.52 (16.54) | *p* = 0.001 * |

* Significant differences, T-Student´s test (*p* > 0.05).

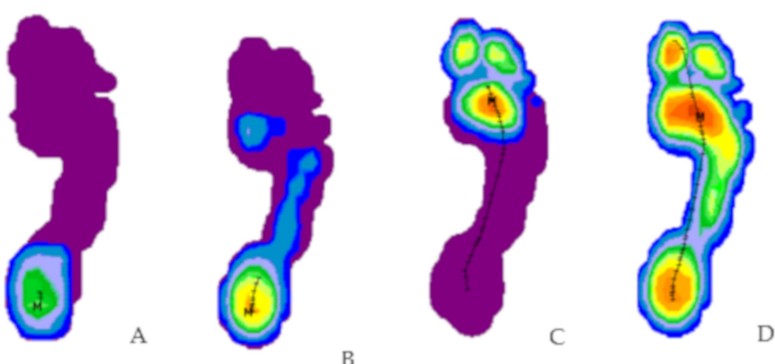

**Figure 1.** *Cont.*

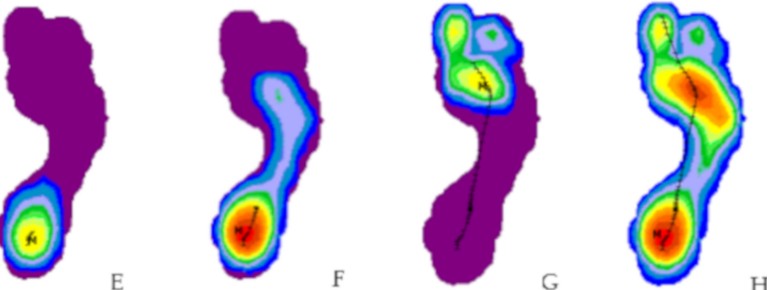

**Figure 1.** Distribution of pressure and surface at 20% of the stance phase of gait for a representative men (**A**) and a representative women (**E**). The black lines show the COP for each group. Distribution of pressure and surface at 35% of the stance phase of gait for a representative men (**B**) and a representative women (**F**). The white lines show the COP for each group. Distribution of pressure and surface at 92% of the stance phase of gait for a representative men (**C**) and a representative women (**G**). The black lines show the COP for each group. Distribution of pressure and surface of global dynamic footprint for a representative men (**D**) and a representative women (**H**).

## 4. Discussion

The results of our study show that, with a functional equinus condition, men have significantly greater plantar contact surface areas than women in all phases of the gait for the first, second, and third steps. Although it was not the aim of this research, a complete study of the reliability variables allows it to be inferred that the first step is the most reliable, and comparing the sex differences of the first, second, and third steps, it is appreciated that the phase of minor gender differences in all steps is the takeoff phase.

The surface values in the contact phase increase in both sexes by increasing the number of steps. Therefore, in the third step, the increase is much higher in men than women. There are previous studies that have examined the plantar footprint in healthy people, where the sole was divided for analysis by anatomical regions or studied as a whole [6,9,10,37–40]. The number and type of anatomical divisions of plantar areas are not consistent across studies. Putti et al. [4] found that an in-shoe system showed an increase of the contact area in 10 parts of the sample. These authors considered that the difference in the contact area could be attributed to differences in BMI. The anatomical analysis separates each part of the foot during the entire stance phase. On the other hand, Murhpy et al. [38] did not find any differences by sex in contact areas in their study of 50 athletes with an in-shoe system. They divided the footprint into four regions. They found a significantly greater value in the area of the rear foot and fore foot in men than women. No significant values were found in the midfoot. The different results between this study and ours could be due to the characteristics of the sample, because athletes participated. Sport activities may change the morphology of the foot [1]. Wunderlich and Cavanagh [37] found the same results as the present study for healthy people. They analyzed gender differences in the shape of the foot and found that men have longer and broader feet than women for any given stature. Male feet differed from female feet in a number of shape characteristics, particularly at the arch, the lateral side of the foot, the hallux, and the ball of the foot. Our study is the first to involve a complete biomechanical analysis focused on sex differences in plantar contact surface areas for the first, second, and third steps. The third step has characteristics of a step in the middle of the gait [40]. We can apply our results to all steps of the gait. Past studies that divided the footprint into anatomical regions took the global dynamic footprint at the end of the step. Then, they separated the heel, midfoot, and forefoot, without taking into consideration the moments where greater forces could be produced. We have chosen the end of each biomechanical phase of the stance phase, because this is where higher velocities of the pressure center occur and the laxity can have a greater influence and generate a greater or equal surface in women than in men. Research has shown greater ligament laxity in women [39]. During the heel strike phase, there are high impact forces due to vertical ground reaction forces [41], and during the mid-stance phase, all the body weight falls on the support leg [41]. Therefore, a greater decrease of the

plantar arch and, consequently, a greater surface area, would be expected in both phases. The study of Hills et al. [11] that compared the third step of the gait in obese adults found a greater surface for women in the mid foot, with anatomical division of the footprint, and found significant increases in pressure under the heel, mid-foot, and metatarsal heads II and IV for men and III and IV for women. Our large sample and biomechanical approach has allowed us to verify that in a functional equinus condition, women present differences during all phases of the stance phase, despite their laxity. These differences remain even in the second and third step, which have more speed [32] and consequently, display greater forces.

Finding sex differences in the selected population is important for research and clinical practice. A functional equinus condition is initially an asymptomatic pathology that causes biomechanical alterations [19], changes in the plantar pressures [19] and tensions and solicitations of the tissues like the Achilles tendon [19]. The findings imply that footwear in a functional equinus condition must take into account this difference in surface, especially sports shoes, and in plantar foot orthoses. The shoes should be narrower and allow greater application in the midtarsal joint, which suffers from excessive pronation. Further studies should be carried out to check if there are also sex differences in children. In the same way, these differences may increase in an older population [42] and therefore, further studies are necessary.

The first step is the most reliable. These results can be explained by the biomechanics of the start of the gait. Prior to the heel contact of the first step, a series of neurophysiological mechanisms and movements are produced. These mechanisms includes a series of stereotyped muscle contractions and inhibitions, invariables of a sequential motor program [43], which is directed at the level of the Central Nervous System, thus unconsciously [43]. The step phase of minor gender differences in all the steps is the takeoff phase and it is the phase where the values are more constant in the three steps. Additionally, it is necessary to consider that the take-off phase is the less reliable phase. The heel strike and mid-stance phase present a moderate to perfect reliability. All variables increase the reliability by using two different days to collect data. These findings can be very useful and be applied in clinical and studies: the first step is the most reliable step to register, and the take-off phase exhibits the least gender differences for the surface variable.

## 5. Conclusions

Significant gender differences in plantar contact surface areas in all phases of the gait with a functional equinus condition were found for the first, second, and third step, and all steps of the gait.

The first step is the most reliable step to register, and the take-off phase is the one with the least gender differences for the surface variable.

**Author Contributions:** E.M.M.-J., R.B.d.B.-V., M.E.L.-I., J.I.D.-V., I.C.-H., C.C.-L., D.L.-L., D.R.-S.: Concept, design, analyses, interpretation of data, and drafting of the manuscript or revising it critically for important intellectual content.

**Funding:** This research received no external funding.

**Conflicts of Interest:** The authors declare no conflicts of interest.

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
