# Peer review of "Sex Differences in the Footprint Analysis During the Entire Gait Cycle in a Functional Equinus Condition: Novel Cross Sectional Research"

_applsci, doi:10.3390/app9173611_

Round 1

Reviewer 1 Report

Comments to the Authors of manuscript number: applsci-577216 entitled Sex Differences of the Footprint Analysis During All Gait Cycle in Functional Equinus Condition: A Novel  Cross Sectional Research.

In the present paper I reviewed the characteristics of footprint in Equinus condition in sex-dependent manner is presented. Equinus is a condition in which the upward bending motion of the ankle joint is limited. And, as it is emphasized by Authors, it is important for research and clinical practice. Depending on how a patient compensates for the inability to bend properly at the ankle, a variety of other foot conditions can develop, such as: arch/heel pain, calf cramping, inflammation of the Achilles tendon, metatarsalgia (pain and/or callusing on the ball of the foot), flatfoot, arthritis of the midfoot (middle area of the foot), pressure sores on the ball of the foot or the arch, ankle pain or shin splints..

The study has been performed in proper manner. Each part of the manuscript is well presented and written, however should correct this manuscript before the acceptance. I recommend minor revision.

The punctuation should be corrected. L 140 – reference should be added. L 182 – reference should be added or the full name of the software. L 211 – not “our sample” but “obtained” L 253 – what is this? – “did n´t” L 279 – this sentence is not finished. L 280 – not “Sex” but “sex” The list of reference should be shortened. The most older should be omitted. And this list should be written in the same manner. English is not my native language, but there are glaring errors. e.g. L 51

Author Response

Dear Reviewer 1:

Kind regards;

The authors

Reviewer 2 Report

Authors investigated sex differences in functional equinus condition.
Particularly, the study focused areas for each of the three phases of stance phase.
Interclass correlation coefficient and standard error of the mean have been used to assess the reliability of the difference.
The conclusion illustrates that males have larger contact surface areas than females during the whole stance phase.

The submitted manuscript was well structured and efficiently present academic findings.
As the authors mentioned the reliability of the first step was higher than the second and third steps were informative.

However, there is one concern regarding the variables to support the conclusion of the study.
Although reasons to extract variables of 20%, 35%, and 92% of the duration of stance phase,
it is not clear if all participant's stance phase reasonably followed the deterministic ratio (i.e. 20%, 35%, and 92%).
As all statistical analysis relies on the extracted variables, better justification of the variables is required.
It would be recommended to show examples that the deterministic ratios are applicable to most or significant participant's stance phase patterns.

Thanks,

Author Response

Dear Reviewer 2,

Kind regards;

The authors.
